# The Behavioural Effects of Innovative Litter Developed to Attract Cats

**DOI:** 10.3390/ani9090683

**Published:** 2019-09-14

**Authors:** Jennifer Frayne, Sarah MacDonald Murray, Candace Croney, Elizabeth Flickinger, Michelle Edwards, Anna Kate Shoveller

**Affiliations:** 1Department of Animal Biosciences, University of Guelph, 50 Stone Road East, Guelph, ON N1G 2W1, Canada; 2Department of Comparative Pathobiology, Department of Animal Sciences, Center for Animal Welfare Science, Purdue University, 625 Harrison St, West Lafayette, IN 47907, USA; 3Kent Pet Group, 2905 N Hwy 61, Muscatine, IA 52761, USA; 4Ontario Agriculture College, The University of Guelph, 50 Stone Road East, Guelph, ON N1G 2W1, Canada

**Keywords:** elimination, attractant, litter, cats, behaviour, video observations, gender

## Abstract

**Simple Summary:**

The global feline population has been rising steadily, with approximately 96.5 million cats in the United States, 8.8 million in Canada, and 102 million in Europe. In addition to a growing population, the complex behaviour of cats results in many being surrendered to animal shelters, especially for urinating and/or defecating outside of the litter box area. The objective of this study was to determine whether cats would demonstrate a preference for a plant-based clumping litter with an added attractant (ATTRACT) over a control plant-based clumping litter (PLANT). A secondary objective was to evaluate whether male and female cats would express different litterbox behaviours as a function of the litter type. Based on video-recorded and live behavioural observations, cats used the ATTRACT litters more often to urinate than the PLANT litter, and males urinated longer and sniffed the litter more than females did (*p* < 0.05). These findings provide preliminary insights into how cat litter-box usage might be increased and gender dimorphisms in litterbox behaviours.

**Abstract:**

Urination and/or defecation outside a designated location has been reported as the most common behavioural reason for surrendering a cat and comprises approximately 30% of cat intakes to shelters. The objective of this study was to determine whether cats would increase in-box elimination when provided a plant-based litter product with attractant (ATTRACT) compared to the same plant-based litter product without attractant (PLANT). Sixteen cats were split into two equal cohorts based on availability from the shelter and group-housed in an enriched room with eight identical litter boxes arranged in a circular pattern equidistant from each other. Following a two-week room acclimation and transition period from clay litter to PLANT litter, boxes were designated either PLANT or ATTRACT litter, balancing for cats’ prior box location preferences. For 14 days following litter allocation, cat behaviours such as sniffing, digging, covering, urinating, and defecating were video recorded for 12 h daily. The cats urinated more often in the ATTRACT litter, suggesting that they preferred the ATTRACT litter for urination more than the PLANT litter (*p* < 0.05). The most significant differences observed were between genders, with males spending significantly more time sniffing and performing urination behaviours (*p* < 0.05). These results suggest that litter with an attractant may be more effective in eliciting usage for urination, as compared to a litter without the additive.

## 1. Introduction

The global feline population has been rising steadily, with household cat numbers reaching approximately 96.5 million in the United States [1], 8.8 million in Canada [2], and 102 million in Europe [3]. These values are consistent with the current human populations of the respective countries. The rise in the domestic cat population and the paucity of information focused on cat behaviour makes optimal management challenging. One indication of this challenge is the large number of cats that enter shelters each year. In the USA alone, 3.2 million cats enter shelters annually [4]. A large percentage of shelter cats are surrendered by their owners, including 30% of all cats entering shelters in Canada [5] and 32% in the UK [6]. While there are many factors that contribute to owner surrender of an animal, elimination outside of the litter box is the number one reason that owners choose to surrender their cat [7,8]. Therefore, efforts focused on understanding elimination behaviour are warranted.

Salman et al. [7] found that 28% of surrendered cats had at least one behavioural problem identified by their owner, and up to 43% of behavioural concerns were urination and/or defecation outside of a designated elimination location. Adopting out a cat with known issues of eliminating outside of the litter box is difficult and often results in the cat being euthanised as the space is needed for more highly adoptable cats with no behaviour issues. According to the Humane Society of the United States, in some areas, up to 70% of cats in shelters may be euthanised [9]. Ideally, if elimination behaviour outside of the litter box is reduced or prevented at home, fewer cats will enter the shelter system.

Eliminating outside of the litter box often reflects distress in cats and is poorly tolerated by most owners [8,10]. While medical issues such as lower urinary tract infection, diabetes, hyperthyroidism, and chronic kidney disease may contribute to the onset of elimination behaviour issues and should therefore be ruled out prior to attempting to address the behaviour [8], determining non-medical causes can be difficult. Cats may eliminate elsewhere in the home for many reasons, but stress is often a factor. Stress may be a result of conflict with other cats in and around the home, recent changes in the home, and problems with the litter box itself [8,11]. Litter box management can be difficult to maintain, and poor or inconsistent management of the litter box is likely a primary reason why cats might start to eliminate elsewhere in the home. Covered litter boxes, boxes that are too small for the cat, providing insufficient litter or infrequent cleaning, and overpowering odors (i.e., too dirty or strong perfumes) can all make cats avoid the litter box [8,12]. 

Previous studies have also shown that cats prefer a small particle size of litter as compared to pellets or pearl particulates and that cats find strong perfumes aversive [13,14]. Clay-based litter is currently the most popular product on the market. However, plant-based litter is an environmentally friendly alternative for owners. It is made from renewable sources, composts much easier, and breaks down faster compared to clay [15]. Furthermore, if the cat or another pet were to ingest the litter, it would be better digested than a clumping clay litter and is therefore considered a safer choice [15]. While good litter management may help ensure that cats use their litter boxes, even with proper management, some cats may still show an aversion to the litter box [10,11]. In addition to the factors previously listed, others such as breed, reproductive status, and age of the cat have been reported to affect elimination behaviour; however, very little is known regarding the role sex might play in these behaviours [16,17,18,19]. Sex is commonly reported as a factor in feline elimination behaviour studies, but it is rarely the main variable tested, and therefore warrants further investigation.

To date, a limited number of studies (<30) have investigated the ability of a litter product to reduce aversion, promote litter box attractiveness and/or increase litter box usage. Cottam and Dodman [20] found that owners reported a significant decrease in the number of eliminations outside of the litter box when using a commercially available cat urine deodoriser to remove the smells of the litter box. Studies have historically focused on adding spray onto the litter to make the litter box more attractive to the cats. Some have reported cats appearing to urinate less often in the box sprayed with an attractant spray on the litter [21], while others showed an increased use of the litter boxes by the cats when the boxes were sprayed with l-felinine (a compound that cats excrete naturally when urinating) [22]. McGlone et al. [23] recently evaluated the effects of pelleted pheromone and interomone additives on aggression near litter boxes in multi-cat homes. The authors reported that one of their compounds (2-methyl-2-butenal) was preferred by the cats over the control, suggesting that adding the compounds directly into the litter might assist in reducing distress and aggression in household cats related to litter-box use [23].

Given the relatively limited number of published studies illustrating the efficacy of litter-box attractants, the objective of the current study was to determine whether cats would prefer a plant-based clumping litter product with an added attractant (ATTRACT) over a control plant-based litter (PLANT). We hypothesised that cats would prefer the ATTRACT litter over the PLANT control, as evidenced by greater amounts of time sniffing, covering, and urinating and defecating. Our secondary objective was to evaluate whether male and female cats expressed different behaviours as a function of the litter type. We hypothesised that males would have a higher rate of preferred elimination behaviour in the ATTRACT litter over the PLANT control litter based on previous literature suggesting that elimination behaviours in cats vary between sexes, with males having a higher prevalence of eliminating outside of the litter box [16,18] 

## 2. Materials and Methods 

The study was conducted according to the guidelines for animal care and use provided by the American Veterinary Medical Association (AVMA), the Canadian Council on Animal Care (CCAC), and the Canadian Veterinary Medical Association (CVMA). All ethical and animal related aspects of the experiment were approved by the University of Guelph Animal Care Committee (AUP#3972). The room itself was approved for cat inhabitation by the Chief Veterinary Inspector of the Ontario Ministry of Agriculture, Food and Rural Affairs (OMAFRA) under the Animals for Research Act prior to the arrival of the cats. 

### 2.1. Subjects

Sixteen healthy cats (seven castrated males, eight spayed females, and 1 intact female), all between 1–5 years in age (mean = 2.25 years ± 0.36), were used for litter preference behaviour observations. All cats in both cohorts were originally sourced from two animal shelters local to the area and had been surrendered by their owners or were picked up in the area as strays. The cats were split into two equal cohorts based on availability from the shelter. Cohort 1 included 3 males and 5 females, while Cohort 2 was an even split between sexes with no significant difference noted between the weights of the females and the males. The age, sex, breed, and colour of each cat was noted upon arrival (Table 1). All cats were group-housed in an indoor room measuring approximately 434 ft^2^. Upon arrival, cats were examined by a licensed veterinarian for general physical issues regarding heart, lung and joint problems and vaccination status was updated if needed. No bloodwork was taken; however, the cats were monitored closely by the caretakers for any possible health complications prior to the study starting. A Feliway^®^ diffuser with Feliway Friends^®^ product was placed in the room to assist the cats with the acclimation to the room and each other for one day prior to arrival and five days after arrival. Cats were provided with vertical climbing and perching surfaces, bedding, hide areas, as well as toys. Cats also had multiple, positive daily interactions with the research team and trained volunteers. The cats participated in voluntary social interactions such as brushing, petting, and playing with the same caretakers two to three times a day during room and litter box cleaning, as well as during as an afternoon session on weekdays, that did not exceed two hours of human presence in the cat room. 

### 2.2. Housing and Enrichment

Cats were given unrestricted access to the entire room, except for the daily 45-min feeding period. Cats were fed once a day in the morning; Cohort 1 was fed at 09:00 h and Cohort 2 at 08:00 h to keep 24-h days consistent during daylight savings time. Each cat was placed in their “condo” (primary housing unit) to allow for exact measurements of feed intake. Each cat condo contained a water dish, hiding box, blanket, and mat. The feeding period allowed for care-taking staff to clean the entire room. The floors were swept and mopped daily, along with the sanitization of all surfaces throughout the room using dilute bleach. Various cat toys, climbing structures, and hiding boxes were provided throughout the room for enrichment. All structures such as the condos, scratching towers, desks, and litter boxes in the room were given a designated position in the room to ensure consistency for the study period. Lighting was maintained on a 12 h light-dark schedule from 08:00 h to 20:00 h. Mean room temperature for Cohort 1 was not recorded properly due to a temperature gauge failure, but the building temperature was recorded daily as approximately 22 °C and mean humidity in the room was 52 ± 4.1%. For Cohort 2, mean room temperature was accurately recorded as 22.17 ± 0.61 °C and mean humidity was 54 ± 11.3%. The volunteer session of interacting with the cats occurred daily from Monday to Friday between 13:00 h and 15:00 h. The litter boxes were scooped twice a day at 08:00 h and 20:00 h with the room being swept each morning.

Eight uncovered litter boxes (Cohort 1 litter boxes were 19¾” × 14” × 5½”, while Cohort 2 litter boxes were similarly sized at 17” × 13½” × 5” but different from Cohort 1 due to availability of the litter boxes) were arranged in a circle in the room (Figure 1). Litter boxes were scooped twice daily and the number of each type of elimination was recorded. The amount of litter in each box was measured in multiple locations to ensure a consistent 2-inch layer was present throughout the study period. All the cats involved in this study had previous exposure to clay-based litter from the shelter; however, it is unknown if any of the cats had been exposed to plant-based litter or if they had any pre-existing behaviour problems with eliminating outside of the litter box. 

Two litters were compared for the study, both being corn product-based products from the same manufacturer. The regular plant-based litter (PLANT) had a loose bulk density of 30.2 lb/ft^3^, particle size distribution of 18.6% of particles retained in the reference rotap screen which were larger than 2.38 mm (ON008) and 8.7% of particles that passed through the reference rotap screen smaller than 0.841 mm (TH020). The test litter had an attractant included in the product (ATTRACT) and had a loose bulk density of 27.1 lb/ft^3^, and particle size distribution of 17.4% ON008 and 6.6% TH020.

For recording purposes, the litter boxes were numbered 1–8, with boxes 1, 3, 5 and 7 containing clay litter and boxes 2, 4, 6 and 8 containing the PLANT litter (Figure 1). The clay litter boxes were transitioned to the PLANT litter by starting out on Day 0 with removing 25% of the volume of the CLAY litter in the morning and adding in the same amount of PLANT litter and mixing the two together. On the morning of Day 2, the volume was calculated to remove enough mixture litter and add in PLANT litter so that there was 50% of the volume in the litter box containing CLAY litter and 50% containing PLANT litter. The removal of mixture and addition of PLANT litter was repeated on Day 4 to ensure that only 25% of the mixture contained CLAY litter and 75% of the mixture was PLANT litter. On the morning of Day 6, all litter boxes containing the mixture were completely emptied and filled with the same amount of litter in the original PLANT litter boxes (litter boxes 2, 4, 6, 8). All eight litter boxes remained with PLANT litter for the remaining 8 days until the ATTRACT litter was introduced. The number of eliminations in each box location was recorded from the four weeks prior to the Attraction period starting and the litter box locations were then ranked. The most used litter box was assigned the PLANT litter, and the second most used litter box was assigned the ATTRACT. This was alternated down the rank list so that four litter boxes held PLANT litter and four held ATTRACT litter.

### 2.3. Behavioural Data Collection 

A video-recording device (Panasonic WV-SPN531) was positioned on the ceiling directly above the litter boxes to allow for simultaneous observation of all eight of them, beginning on day 00. The litter box area was recorded for 12 h continuously during the light period from 08:00 h to 20:00 h, for all 28 days of the study, and for each cohort. The data were analyzed from days 15 to 28, which corresponded with the placement of plant-based litter with and without the attractant. Video analysis was performed by two observers and tested for inter-observer reliability. The total number of eliminations for each litter box was recorded twice a day, at morning and evening litter box scooping. Any eliminations outside of the litter box and locations also were recorded.

### 2.4. Ethogram and Behaviour Coding

The ethogram used for this study was based on the one designed by McGowan et al. [11], which consisted of 39 different behaviours. This ethogram included 10 primary behaviours based on litter box involvement only; four of which were subdivided into more detailed categories to discriminate among pre-box use event, during box use, and post-box use event (Table 2). The duration of each coded behaviour was determined by marking the time when the cat first started litter box-oriented behaviour and when the cat stopped the behaviour. In addition to recording each behaviour occurrence, the time, duration, and location of the litter box that was used were noted. To be considered an event, there must have been a minimum of one behaviour observed that was listed in the ethogram. An event was considered concluded when the cat left the litter box area.

### 2.5. Inter-Observer Reliability

Two independent observers were used to record each cohort for 12 h a day for 28 days. The two observers were evaluated for inter-observer reliability with the same five days, resulting in a β-score above 0.75. Referring to Bakeman and Gottman’s earlier work [24], the coders were considered to have excellent reliability characterizations and could code each cohort independently.

### 2.6. Statistical Analyses

Data were analyzed using PROC GLIMMIX in SAS (SAS Statistical Software 9.4 Release M4, SAS Institute Canada, Toronto, ON, Canada) and modeled either using a Lognormal response distribution (sniff frequencies and duration) or a Poisson distribution (defecation and urination frequencies and durations). Results were statistically significant at *p* < 0.05 level and trends were identified at *p* < 0.10 level. Statistical analysis of the ethogram coded data was performed using SAS (SAS Statistical Software 9.4 Release M4, SAS Institute Canada, Toronto, ON, Canada) and was primarily used to look at the impact of treatment or sex on feline litter behaviour frequency and duration. When the model did not result in normally distributed residuals, the appropriate linear and/or lognormal distribution model was used for analysis. Frequency, total duration, and duration of each behaviour were analyzed based on cat sex and litter type. Data were evaluated according to main effects and interactions of treatment and sex, which included subclasses identified in the ethogram (Table 2). The total duration was defined as the total amount of time spent performing the respective behaviour for each treatment and sex category used over the entire study period analyzed (14 days). Duration was defined as the length of time (in seconds) a cat was observed performing a behaviour. 

## 3. Results

Each cohort was analyzed separately as a fixed effect in our model, but the cohort was not found to be significant, so the results of both cohorts were pooled and analyzed together. Throughout the 14-day trial period for both cohorts, all sixteen cats were observed covering, yielding a total of 209 observations. Digging behaviour was observed 110 times from seven males and eight females, and mimic behaviour was observed a total of 57 times, from three males and five females. The trial period included 242 urination events from all 16 cats, in addition to 114 defecation events, which led to 368 occurrences of sniffing behaviour. Cats were found to have a positional preference for the number 3 litter box spot, which was the only litter box next to the wall in the circle, and they also appeared to prefer to use the litter box immediately after scooping. By adding up the number of total elimination events (urination and defecation), it was found that the cats in each cohort both used this spot more often than any other litter box. Upon video analysis by the coders or visual live observations by the caretakers, it was found that no eliminations were noted outside of the litter box during this study period. 

Numerical means with appropriate standard deviations and corresponding *p*-values for the counts, total durations, and durations for each behaviour are listed in Table 3 and Table 4. 

### 3.1. Cover Behaviour

During the observation period, neither sex nor litter type significantly affected cover behaviour count (*p* > 0.10). Males covered significantly longer than females over the 14-day trial period (*p* = 0.0375). Daily cover behaviour duration observed in males also tended to be longer than females (*p* = 0.0626). Cover duration was not significantly affected by litter type (*p* > 0.10) (Table 3).

### 3.2. Dig Behaviour

Neither sex nor litter type significantly affected dig behaviour frequency (*p* > 0.10). Total duration of dig behaviour throughout the entire 14-day period was not significantly affected by sex (*p* > 0.10). Daily duration of dig behaviour was not significantly affected by sex or litter type either (*p* > 0.10) (Table 3).

### 3.3. Mimic Behaviour

During the observation period, neither sex nor litter type significantly affected mimic behaviour frequency (*p* > 0.10). Total duration of mimic behaviour throughout the entire 14-day period was not significantly affected by sex nor litter type (*p* > 0.10). Daily duration of mimic behaviour was not significantly affected by sex or litter type (*p* > 0.10) (Table 3).

### 3.4. Sniffing Behaviour

No differences were found when the sniff-pre and sniff-post behaviours were analyzed separately (*p* > 0.10). All sniff behaviours (sniff, sniff-pre-and sniff-post) were analyzed together, by modelling a Lognormal response distribution for frequency and total study duration of sniff behaviours. When the data for both sexes (male and female) were combined, there were no differences reported between the ATTRACT and PLANT litter for sniffing behaviour in terms of number of times the behaviour occurred, as well as duration of the behaviour (*p* > 0.10). When adding sex as a fixed effect to determine the difference in sniffing behaviour between males and females, the male cats sniffed significantly more frequently per day than the females (Males 5.24 ± 0.77 times; Females = 3.49 ± 0.45 times; df = 352, *p* = 0.0001). This is consistent with total sniffing duration for the entire study period as males consistently sniffed more than females (Males 69.97 ± 11.42 s; Females = 30.38 ± 4.410 s; df = 350, *p* = 0.0002) (Table 4).

### 3.5. Urination Behaviour

Data were analyzed by modelling a Poisson response distribution for count and daily duration of urination behaviour and Lognormal response distribution for total duration of urination behaviours. In general, the cats were found to urinate significantly more in the ATTRACT litter than the PLANT litter over the study period (ATTRACT = 6.34 ± 0.50 times; PLANT = 5.63 ± 0.46 times; df = 224, *p* = 0.0285) but no significant differences were found between the number of urinations in males and females per day (*p* > 0.10). Duration of urination behaviour did not differ between the litter types; however, males were found to urinate significantly longer each day than females (MN = 22.02 ± 2.14 s; FS = 11.28 ± 0.99 s; df = 224, *p* < 0.0001). This was consistent across the entire study period, as total duration was higher for males than females (MN = 96.53 ± 10.430; FS = 54.13 ± 5.10; df = 226; *p* < 0.0001) (Table 4).

### 3.6. Defecation Behaviour

Data were analyzed by modelling a Poisson response distribution for frequency and daily duration of defecation behaviour and Lognormal response distribution for total duration of defecation behaviours. There were no significant differences found between the number of defecation behaviours between the different litters or between the sexes. Duration of defecation behaviour did not vary between the sexes either on a daily or total study period. However, it was found that the cats generally had a longer daily defecation duration in the ATTRACT litter than the PLANT litter (ATTRACT = 29.51 ± 1.83 s; PLANT = 27.14 ± 1.65 s; df = 96, *p* = 0.035) (Table 4).

## 4. Discussion

This is the first study to examine whether a plant-based litter additive can alter feline elimination behaviour. By using frequency, total duration and duration of six behaviours previously associated with litter box use by McGowan et al. [11], we were able to investigate the preference of cats using a litter product with attractant. Cats urinated more often in the ATTRACT litter than the PLANT, suggesting that cats preferred the ATTRACT litter for urination more than the PLANT control. In this study, male cats spent more time sniffing and performing urination behaviours than females (Table 4). Previous research by Barry and Crowell-Davis [17] hypothesised that this increased duration of sniffing in neutered male cats is a residual trait from mating behaviour. This is consistent with what was found in the current study where males displayed both behaviours for significantly longer than females, and males were found to urinate longer than females each day, as well as throughout the whole study.

Four of the six behaviours were observed separately and distinguished between pre-elimination, during elimination, and post-elimination, suggesting a behaviour sequence by cats when interacting with a litter box [11]. No differences were noted during any sniffing behaviour (pre-, during, post− or combined), and olfactory communication within a species has previously been found to be an important behaviour. However, olfaction behaviour in cats is a complicated behaviour that requires further investigation, particularly regarding cats’ perceptions of and responses to different odors [10]. Cats often scent mark either via urine or glands in their face and paws [25]. However, urinary scent marking in the room was not observed by the caretakers or video-coders in the present study.

The effects on elimination behaviour of having multiple cats in one room must be considered. The dynamics that occurred between cats, especially resource guarding, could certainly have influenced the behaviour of how the cats interacted with the litter boxes. In multiple cat households, resource guarding, such as blocking conspecifics from use of a litter box or even defensive aggression around a resource, are often reported [26,27]. Behaviors that appeared to be resource-guarding incidents around the litter boxes included chasing conspecifics and were noted less than 10 times for both cohorts while reviewing the videos. However, more detailed analysis of the video footage is needed to determine if these were indeed resource guarding and associated with similar behaviours around various other resources besides the litter boxes (such as food, toys, and resting areas). The study design itself did assist in reducing the chances of resource guarding in the room by having multiple water bowls, toys, climbing structures, and enough litter boxes for each cat. While each cohort was acclimated to the room for at least two weeks to allow the cats to become accustomed to each other and establish a hierarchy, multi-cat interactions can be a complicated process of events and can add additional stress to the cats, resulting in changes in behaviour [27,28,29]. This also could have potentially led to variations in cat elimination behaviour due to aggressor cats guarding the litter box area [29]. We also observed that the cats had a positional preference for litter boxes, with the most favoured position being against a wall so that they can watch their surroundings at the same time as using the litter box [30]. Litter box frequency use was ranked by the cats’ preferences for litter box locations in the acclimation and transition phases and alternated to reduce cat preference bias. The PLANT litter was placed in the cats’ most preferred position against the wall to eliminate any confounding results of the ATTRACT litter in that position. The cats were often observed eliminating immediately post-scooping of the litter boxes, confirming that litter cleanliness plays a critical role in elimination behaviour [31].

Cat behaviour is inherently variable between individuals and dependent on genetics and the environment, including the early rearing environment. The authors recognise that with a different group of cats, the outcomes to the study may have differed. Limitations of this study therefore include lack of knowledge about previous litter box behaviour in the cats, as they were from an animal shelter. The history of behavioural problems was relatively unknown, even for cats who were surrendered to the shelter, due to the possibility of inaccurate information being provided by previous owners surrendering their cats. Another limitation is that cats housed in cage environments, as shelter cats usually are, may be distressed, leading to different behavioural patterns and problems. The added stress of being housed in an animal shelter could have potentially pre-exposed the cats to certain behaviours prior to arrival at the research location [32,33]. Being in a novel environment and group-housing random-sourced cats can also change behaviour in cats and result in resource-guarding and defensive behaviour [29]. To account for these stressors in the study design, the cats were acclimated longer than usual in research settings (typically one week) and they were also given a longer transition period to acclimate to other cats in the room [34]. Young adult cats were chosen for the study, as it was assumed that they would have a pre-existing preference to clay litter. The cats’ pre-exposure to clay litter was assumed based on current market products available for cat litter and because clay litter is typically offered in animal shelters. However, it is important to note that the preferences for litter may be different for older cats as the roles of experience and behavioural flexibility of older cats have not been adequately studied relative to litter usage. An established colony of cats with no history of eliminating outside of the litter box would help to remove any confounding effects of prior experience of eliminating out of the litter box and litter preference. However, since such a model was not available and would not reflect real-life conditions, shelter cats were used. In addition to age and prior experience, other factors such as breed have been known to contribute to the risk of displaying elimination behaviour. Further, the effect of the sex of the cat on litter box behaviour has been suggested as a potential factor influencing litter box use and should be examined further [16,17,18,19].

Understanding the behaviour of cats regarding ensuring consistent litter box usage remains a challenge. However, the use of an attractant in plant-based litter, along with other well-established litter box management strategies, may help reduce the number of cats exhibiting house soiling [34,35,36]. This is particularly important to support cat welfare and the human-cat bond, as owner reports of house soiling are increasing worldwide, along with increased owner surrenders to animal shelters [1,5,6], which may be related to intolerance of house-soiling. Learning more about the needs and behaviour of cats may provide a platform to help educate owners about the role of litter and litter-box maintenance in encouraging appropriate urination and defecation behavior, and ultimately improving the human-animal bond.

## 5. Conclusions

Overall, the results of this study both add to the basic understanding of feline elimination behaviour and suggest potential targets for improving litter preference and reducing house soiling. When analyzing the data from both Cohorts, cats urinated more in the ATTRACT litter than the PLANT litter. We propose that more urination events are a sign of preference, and therefore suggest that cats preferred the ATTRACT litter over the control. When used in conjunction with good litter box management, the ATTRACT litter may help to ensure that cats continue to be drawn to their litter boxes for elimination, thus helping to reduce the likelihood of urinating outside of the box.

Each cat exhibited unique behaviours, suggesting that elimination behaviour is far more complex than previously hypothesised. Cat elimination behaviour has been shown to be affected by numerous biological and environmental factors that may not have been previously considered. Among the six behaviours observed, the most significant differences between sexes were found for the duration of sniff and urination behaviours, suggesting fundamental physiological differences between male and female cats.

Given the differences between male and female elimination behaviours reported in this study, future research should focus on identifying the litter box behaviour differences between male and female cats, to learn more about risk factors for eliminating outside of the litter box.

## Figures and Tables

**Figure 1 animals-09-00683-f001:**
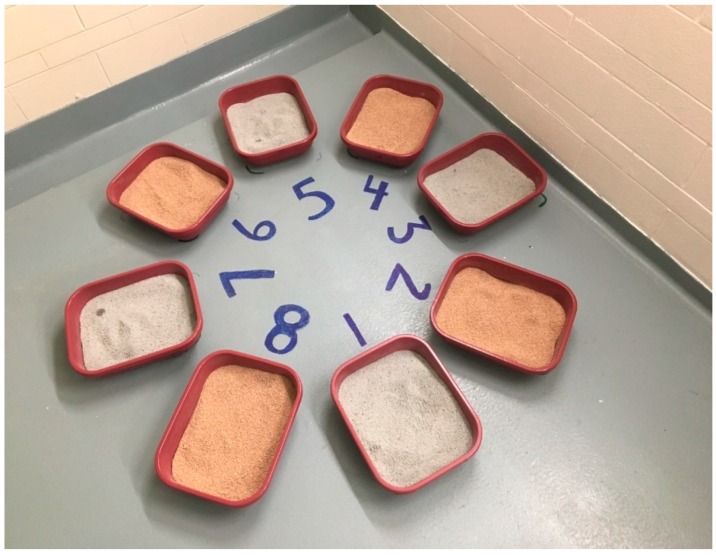
The circular arrangement of litter boxes at the start of the trial with Cohort 2 containing alternating clay and plant-based litter substrates.

**Table 1 animals-09-00683-t001:** Cat demographics by cohort.

	Cat Name	Gender	Age (Years)	Breed	Colour
Cohort 1	Brownie	F	1	DSH	Brown tabby
	Elton John	M	5	DSH	Black
	Scarlet	F	1	DSH	Black/White
	Eggbert	M	2	DSH	Brown tabby
	Simmy	F	1	DSH	Brown tabby
	Leah	F	5	DSH	Orange tabby/White
	Riley	M	3	DLH	Orange
	Rosalie	F	3	DLH	Orange
Cohort 2	Gretchen	F	3	DSH	Black/White
	Violet	F	3	DLH	Black/White
	Betsy	F	1	DSH	Tortishell
	Mildred	F	1	DSH	Brown tabby
	Norbert	M	1	DSH	Grey tabby
	Larry	M	1	DLH	Grey
	Tim	M	1	DSH	Brown/White
	Leo	M	2	DSH	Orange tabby

Abbreviations: F–Female; M–Male; DSH–Domestic short-hair; DLH–Domestic long-hair.

**Table 2 animals-09-00683-t002:** Ethogram detailing the recorded behaviours.

Behaviour	Description
Covering	Partial or complete covering of the cat’s own elimination through displacement of litter by their paws post-elimination.
	Partial or complete covering of an elimination that does not belong to the cat through displacement of litter by their paws.
Digging	Displacement of litter by a cat’s paws prior to elimination or without elimination.
Fecal event	Begins when complete posture is taken and ends once the stance is broken. Confirmed by fecal presence in the litter box.
Mimic covering	Covering behaviour where the floor is scratched/hit in place of litter.
	Covering behaviour where the litter box is scratched/hit in place of litter.
	Covering behaviour where the wall is scratched/hit in place of litter.
Paws in the litter box	1 paw in the litter box.
	2 paws in the litter box.
	3 paws in the litter box.
	4 paws in the litter box.
Perching	2 + paws resting on the edge of the litter box.
Posturing	Cat positioning for elimination characterised primarily by the upwards slanting of the back.
Sitting	The cat is sitting in a litter box.
Sniffing	Presence of the cat’s head over the litter box or their nose at the edge of the box that is not associated with an elimination.
	Presence of the cat’s head over the litter box or their nose at the edge of the box immediately prior to elimination in that box.
	Presence of the cat’s head over the litter box or their nose at the edge of the box immediately following an elimination in that box.
Urination event	Begins when complete posture is taken and ends once the stance is broken. Confirmed by urine presence in the litter box.

**Table 3 animals-09-00683-t003:** Least square means (± Standard Deviations (SD) for frequency, total duration and duration of covering, digging and mimicking behaviour.

**Covering**											
	**Attract**	**Plant**	***p*-Value**	**Female**	**Male**	***p*-Value**	**Female Attract**	**Female Plant**	**Male Attract**	**Male Plant**	***p*-Value**
Frequency	1.45 ± 0.10	1.48 ± 0.10	0.7799	1.36 ± 0.10	1.58 ± 0.10	0.1069	1.37 ± 0.11	1.35 ± 0.10	1.54 ± 0.11	1.61 ±0.12	0.5874
Total Duration	-	-	-	11.30 ± 2.58	22.20 ± 5.04	0.0375	-	-	-	-	-
Duration	10.88 ± 1.73	9.63 ± 1.50	0.3389	7.81 ± 1.61	13.41 ± 2.71	0.0626	7.98 ± 1.85	7.65 ± 1.69	14.84 ± 3.24	12.11 ± 2.67	0.5284
**Digging**											
	**Attract**	**Plant**	***p*-Value**	**Female**	**Male**	***p*-Value**	**Female Attract**	**Female Plant**	**Male Attract**	**Male Plant**	***p*-Value**
Frequency	1.27 ± 0.12	1.31 ± 0.14	0.7629	1.17 ± 0.14	1.42 ± 0.13	0.2079	1.17 ± 0.15	1.17 ± 0.21	1.38 ± 0.15	1.47 ± 0.15	0.8014
Total Duration	-	-	-	NBV *	NBV *	NBV *	-	-	-	-	-
Duration	9.79 ± 1.43	11.62 ± 2.27	0.4341	9.82 ± 2.23	11.58 ± 1.67	0.5402	8.99 ± 2.12	10.72 ± 3.79	10.67 ± 1.84	12.57 ± 2.11	0.9780
**Mimicking**											
	**Attract**	**Plant**	***p*-Value**	**Female**	**Male**	***p*-Value**	**Female Attract**	**Female Plant**	**Male Attract**	**Male Plant**	***p*-Value**
Frequency	1.43 ± 0.18	1.29 ± 0.15	0.4932	1.30 ± 0.15	1.43 ± 0.21	0.6012	1.28 ± 0.20	1.31 ± 0.18	1.60 ± 0.32	1.28 ± 0.23	0.4234
Total Duration	-	-	-	26.93 ± 15.75	45.14 ± 20.57	0.4855	-	-	-	-	-
Duration	50.04 ± 13.78	32.45 ± 12.62	0.3684	27.48 ± 11.37	55.01 ± 14.82	0.1714	25.18 ± 16.17	29.79 ± 15.99	74.91 ± 23.31	35.11 ± 19.52	0.2620

* NBV = No back-transformed value.

**Table 4 animals-09-00683-t004:** Least square means (± SD) for count, total duration and duration of urination, defecation and sniff behaviour.

**Urination**											
	**Attract**	**Plant**	***p*-Value**	**Female**	**Male**	***p*-Value**	**Female Attract**	**Female Plant**	**Male Attract**	**Male Plant**	***p*-Value**
Frequency	6.34 ± 0.50	5.63 ± 0.46	0.0285	6.21 ± 0.62	5.74 ± 0.65	0.5988	6.41 ± 0.67	6.03 ± 0.64	6.27 ± 0.74	5.25 ± 0.65	0.2852
Total Duration	-	-	-	54.13 ± 5.10	96.53 ± 10.43	<0.0001	-	-	-	-	-
Duration	15.63 ± 1.05	15.89 ± 1.08	0.6377	11.28 ± 0.99	22.02 ± 2.14	<0.0001	11.56 ± 1.05	11.01 ± 1.02	21.14 ± 2.10	22.94 ± 2.30	0.0579
**Defecation**											
	**Attract**	**Plant**	***p*-Value**	**Female**	**Male**	***p*-Value**	**Female Attract**	**Female Plant**	**Male Attract**	**Male Plant**	***p*-Value**
Frequency	2.83 ± 0.43	2.94 ± 0.43	0.7399	2.98 ± 0.55	2.79 ± 0.56	0.8138	3.33 ± 0.67	2.66 ± 0.54	2.40 ± 0.55	3.25 ± 0.67	0.0291
Total Duration	-	-	-	65.29 ± 10.13	71.86 ± 11.91	0.6735	-	-	-	-	-
Duration	29.51 ± 1.83	27.14 ± 1.65	0.0353	26.54 ± 2.09	30.18 ± 2.60	0.2727	27.52 ± 2.32	25.59 ± 2.14	31.64 ± 2.88	28.78 ± 2.55	0.7767
**Sniffing**											
	**Attract**	**Plant**	***p*-Value**	**Female**	**Male**	***p*-Value**	**Female Attract**	**Female Plant**	**Male Attract**	**Male Plant**	***p*-Value**
Frequency	4.37 ± 0.46	4.18 ± 0.44	0.5728	3.49 ± 0.45	5.23 ± 0.77	0.0401	3.36 ± 0.47	3.63 ± 0.51	5.68 ± 0.90	4.82 ± 0.76	0.1264
Total Duration	-	-	-	30.37 ± 4.39	70.01 ± 11.39	0.0001	-	-	-	-	-
Duration	45.47 ± 6.35	46.75 ± 6.58	0.8751	30.38 ± 4.41	70.00 ± 11.42	0.0002	27.67 ± 5.13	33.36 ± 6.26	74.72 ± 15.60	65.51 ± 13.75	0.3660

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
