# Peer review of "The Behavioural Effects of Innovative Litter Developed to Attract Cats"

_animals, 2019, doi:10.3390/ani9090683_

Round 1
Reviewer 1 Report
The longer we insist on using the term inappropriate elimination, the longer we will have less chance of resolving the issue. Elimination is not inappropriate and when we say this, we imply that a cat is doing something wrong. If we call the behavior by a descriptive title such as house-soiling or elimination outside of a litter box we don't imply that the cat is mis-behaving. Consequently the owner/care-give may view it as a condition that veterinary behaviorists can positively influence. If 2 sizes of litter boxes were available would mixing sizes between the cohorts be less likely to possibly have a size preference implication? Yes, confined cats are more likely to use any LB offering until it is really too small but a mixture would be more representative of what a cat in a home situation might encounter. Did any of the cats in the study eliminate outside the LB at any time during the study? I found the statement about spray-marking not occurring but not one about all forms of soiling.
Author Response
Please see attachment for response to reviewer comments

Reviewer 2 Report
I think this study needs to be clearly flagged as an N=2 study, and this needs to be strongly reflected in the limitations and discussion. It is N=2 because each group is composed 8 of non-independent individuals.
L20 and 51: 102 million cats in UK? Obviously can't be correct
L21: a growing population (check throughout for tautologies)
L22: delete "inappropriate elimination which is"
Why ATTRACT and PLANT and not just P and P+.. seems rather a long alternative
L57: A large percentage of "these cats". What are "these cats"? Especially since the comparisons are between US and 2 other countries.
L61: surely "one behaviour of concern", a cat cannot be concerned about its own behaviour.
L64: reference concerning IE being a cause of euthanasia? Does it really prevent rehoming or does it increase post-adoption relinquishment?
L86: reference? If this is a postulation it needs to be marked more clearly.
L91: if there are a limited nyumber state how many there are (is it 1 or 2 or 10-12).
L92: delete "one such study by" as it is unnecessary
L96: attractive?
L110-111: Why hypothesise that males do more? Is this post hoc? Seems an odd a priori assumption and requires a reference to demonstrate the logic.
124: was the elimination behaviour of the cats known? As you say in the introduction IE is a major cause of relinquishment. You later indicate tbhis was unknown, it is therefore a major limitation. Was IE not recorded for each cat in a baseline observation experiment?
L126: Animals have a sex not a gender (gender is a human socio-biological construct)
L128: What did the vet find/look for?
L134: I assume these individuals were the human caretakers?
L171: what do ON008 and TH020 mean?
L186: starting not started
Method: I am a bit concerned that the cats had extensive experience of the P litter but none of the P+ prior to the data recording. This means that any difference in useage couild be enmtirely attributed to novelty and not "attraction" per se.Why did the authors not follow the same protocol for P+ (for example by performing the replacement protocol in reverse between pens 1 and 2). This would be: Pen 1: start with clay and P and replace clay with P+; Pen 2: start with Clay and P+ and replace clay with P.
Why were data not collected during the transition period from clay to plant material? This would give some indication of preference for the new litter before the project beginning. It would also provide a baseline of IE
L194: The data were analysed (please treat "data" as a plural throughout)
L203: "between" refers to 2 categories, if looking at more it is "among". Pre-box and post-box are odd terms (a pre-box would be something used before the box). It chould be "events prior to/following litterbox use"
Mimic covering: "of" not "if" for 3rd category
L231: cohort is not a proper noun so does not need to be capitalised.
Why is the number of cats constantly repeated? Why not "all 16 cats displayed behaviours X Y Z"
Table 3: Why P-value to 4sf? I would use 2
255-264: The 9 lines can simply be reduced to: "There were no sig diffs in mimic or dig behaviours between the sexes or litter types" (or similar).
L266: again sniff-pre and sniff-post are rather odd terms, just say "before" and "after"
section 3.4: Do male cats not display more sniffing anyway based on their inate biology? especially if females are urinating in the same trays.. and even more so if one of the females is unsterilised.
L284: were males significantly larger and therefore had larger bladders? Given number of urinations doesn't differ it would suggest so.
L281: is the difference between 6.34 and 5.63 really of biological significance. Should these data not be presented as median values as you cannot have a partial urination? Likewise is a variation in length of defaecation of 1.5s really of value?
I feel that there has been an over analysis of the data and the findings may be type-errors, why compare events, total durations and event durations?
Why are the IE data not presented? How many were there over the study? Did IE reduce with the removal of the clay litter (which would allow for preference between clay and plant litters to be assessed. Did cats with IE improve over the study? These are surely some major values in any litter given the introductions recognition of IE as a major issue.
L301: as earlier, I think a difference between the 2 treatments of ~0.5 urinations in n=2 cohorts means such an emphatic statement is unwarranted. I would even go so far as to say that the differences were inconclusive.
L394: do residual and vestigial not mean the same thing?
L337: should these ideas not be in the results section before introducing discussion of them? New results shouldn't appear in the dicsussion.
Can behaviour in a multi-cat cohort in a relatively invariant space really be related to in-home experiences?
Author Response

(The authors gave the same response as above.)
